# Radiolabeled Iron Oxide Nanoparticles as Dual Modality Contrast Agents in SPECT/MRI and PET/MRI

**DOI:** 10.3390/nano13030503

**Published:** 2023-01-27

**Authors:** Maria-Argyro Karageorgou, Penelope Bouziotis, Efstathios Stiliaris, Dimosthenis Stamopoulos

**Affiliations:** 1Department of Physics, National and Kapodistrian University of Athens, 15784 Athens, Greece; 2Institute of Nuclear & Radiological Sciences & Technology, Energy & Safety, National Center for Scientific Research “Demokritos”, 15341 Athens, Greece; 3Institute of Nanoscience & Nanotechnology, National Center for Scientific Research “Demokritos”, 15341 Athens, Greece

**Keywords:** iron oxide nanoparticles, magnetic resonance imaging, radionuclides, single photon emission computed tomography, positron emission tomography, radiolabeling, dual modality contrast agent, passive targeting, selective targeting, diagnosis

## Abstract

During the last decades, the utilization of imaging modalities such as single photon emission computed tomography (SPECT), positron emission tomography (PET), and magnetic resonance imaging (MRI) in every day clinical practice has enabled clinicians to diagnose diseases accurately at early stages. Radiolabeled iron oxide nanoparticles (RIONs) combine their intrinsic magnetic behavior with the extrinsic character of the radionuclide additive, so that they constitute a platform of multifaceted physical properties. Thus, at a practical level, RIONs serve as the physical parent of the so-called dual-modality contrast agents (DMCAs) utilized in SPECT/MRI and PET/MRI applications due to their ability to combine, at real time, the high sensitivity of SPECT or PET together with the high spatial resolution of MRI. This review focuses on the synthesis and in vivo investigation of both biodistribution and imaging efficacy of RIONs as potential SPECT/MRI or PET/MRI DMCAs.

## 1. Introduction

In a clinical setting, various medical imaging techniques are employed to visualize diseases, such as cancer. Each of these techniques possesses unique strengths and limitations [1,2], so they differ in terms of their capabilities and subsequent use in clinical practice. For example, referring to their capabilities, magnetic resonance imaging (MRI) and computed tomography (CT) are known to provide high spatial resolution, namely 25–100 μm and 50–200 μm, respectively, yet they are not as sensitive as the nuclear imaging techniques, such as single photon emission computed tomography (SPECT) and positron emission tomography (PET) [3]. In fact, the sensitivity of MRI ranges between 10^−9^ and 10^−6^ mol/L and that of CT is approximately 10^−6^ mol/L [3]. On the other hand, the nuclear imaging techniques are well known for their high sensitivity, since PET can detect chemical compounds in concentrations ranging between 10^−11^ and 10^−12^ mol/L, while SPECT detects slightly higher concentrations of the compounds, which range between 10^−10^ and 10^−11^ mol/L. However, both SPECT and PET provide low spatial resolution of 1–2 mm [3]. Referring to their use in daily clinical practice, in general, MRI and CT are employed to provide information on anatomical abnormalities of a particular disease, whilst both SPECT and PET are able to identify the functional/biochemical changes, which are related to the respective disease and most of the time, they precede the ones of the anatomical abnormalities [3]. However, in the recent decades functional MRI has also been used in clinical practice for brain imaging, while hyperpolarized carbon-13 based MRI is additionally used in clinical trials for probing timely and accurately the metabolic changes that are associated with diseases such as cancer, cardiovascular disease, hepatic and renal disorders, etc. Thus, these advanced versions of MRI can offer information that is directly compared to the one provided by PET and/or SPECT [4,5]. Therefore, combining different imaging techniques into one modality is useful, because in this way the limitations concerning each imaging technique can be surpassed, allowing for the provision of synergistic information.

Among the various magnetic materials used in diagnostic applications, iron oxide nanoparticles (IONs), specifically magnetite, Fe_3_O_4_, and/or maghemite, γ-Fe_2_O_3_, have drawn a lot of attention for their biocompatibility [6,7,8,9,10], biodegradability [11], and easy synthesis [12]. Additionally, IONs have the propensity to exhibit superparamagnetic properties at room temperature below a certain size, that is, lower than 20 nm for Fe_3_O_4_ and lower than 30 nm for γ-Fe_2_O_3_ [13]. In the superparamagnetic state, the IONs maintain their colloidal stability because they do not indicate remanence magnetization upon the absence of an externally applied magnetic field. The absence of aggregates is crucial for biomedical applications, since it reduces the risk of capillary vessels embolism. Moreover, it is well-known that the aggregates are taken up via phagocytosis by the macrophages of the organs of the reticuloendothelial system (RES), ultimately leading to their fast elimination from the bloodstream [13]. For these reasons, the FDA has approved the clinical application of IONs in diagnosis as MRI contrast agents. In particular, IONs with sizes more than 20 nm (known as superparamagnetic IONs, namely SPIONs) are referred to as negative contrast agents in MR imaging, while the ones with sizes less than 5–10 nm (knows as ultrasmall SPIONs, namely USPIONs) are referred to as positive MRI contrast agents. This is because they both affect the spin-lattice (T_1_) and the spin-spin (T_2_) relaxation times of the nearby protons found mainly in water molecules. As a result, they produce either hypointense (in the case of the negative contrast agents) or hyperintense (in the case of positive ones) signals that give prominence to the organs or tissues in which the IONs have accumulated [2]. Various diagnostic MRI studies, including imaging of cancer [14], lymph nodes [15], Alzheimer’s disease [16], etc., have reported their use as MRI contrast agents. However, in the case of the negative contrast agents, sometimes the produced hypointense signal cannot be efficiently distinguished by the signal that is produced by endogenous conditions, like calcification, hemorrhage, and blood clots.

Therefore, in the ideal case, MRI and IONs should be empowered by additional imaging techniques and contrast agents of different functionality, thus having the advantage to provide supplemental information. For instance, the simultaneous application of MRI and nuclear imaging (SPECT or PET) combines the advantages of each imaging technique, hence providing high quality hybrid images which ultimately lead to a timely and more accurate diagnostic effect for the benefit of the patient. Consequently, over the last few decades, radiolabeled iron oxide nanoparticles (RIONs) have emerged as potential dual modality contrast agents (DMCAs) in various imaging applications. To become a DMCA, the IONs are radiolabeled with various diagnostic radionuclides already used in clinical practice [17].

Many studies have referred to the development and in vivo investigation of the biodistribution and imaging efficacy of RIONs as potential SPECT or PET and MRI DMCAs for passive or selective imaging of diseases. Hence, the aim of the present review is to provide a clear and informative guide on passive/selective RIONs, as we believe that they could represent a promising tool for early diagnosis.

## 2. IONs as MRI Contrast Agents

The MRI takes advantage of the tissues’ T_1_ and T_2_ relaxation times or their density in protons in order to produce the MRI image, albeit in a clinical setting, the contrast of the image is not always adequate. Therefore, the discrimination of a tissue under study (i.e., tumor) from its environment is inadequate, resulting in inefficient diagnostic interpretation of the corresponding disease. For this reason, the use of MRI contrast agents is very important, as they can improve the contrast on MR images and thus the overall diagnostic outcome. As it has previously been mentioned, IONs are employed as contrast agents in MRI. Due to their ability to produce local inhomogeneities in the externally applied magnetic field, the presence of IONs ultimately alter the T_1_ or T_2_ relaxation times of nearby protons. As a result, they alter the signal intensity in the MR image and provide either positive or negative contrast enhancement in the tissues where they accumulate [2,18,19,20]. Thus, a few years ago, several MRI contrast agents based on IONs have already been applied in clinical diagnosis, including Feridex IV and Resovist for the diagnosis of malignancies in the liver [2,12,18], as well as Lumiren and Combidex (or Sinerem) for the inspection and imaging of the bowel and lymphatic node metastases, respectively [12,18].

In general, IONs (SPIONs and USPIONs) are considered biocompatible enough, and this is the main reason why they are used worldwide in daily clinical practice [11,13,18,21]. Specifically, after the degradation of IONs by the lysosomes of macrophages, they are naturally stored as iron cations into the body via ferritin in order to be re-used in vital biochemical processes, such as the formation myoglobin and hemoglobin [13,21]. However, it should be noted that in some cases (i.e., in case of iron overload), IONs can cause adverse effects via the induction of oxidative stress elicited by the production of reactive oxygen species (ROS) through the Fenton reaction [13]. For this reason, many studies have evaluated the toxicological profile of the IONs, which is dependent on many factors, such as their physicochemical characteristics (i.e., size, charge, coating material, etc.), the type of cells studied, route of administration, and the concentration of the injected dose [13,21].

In addition to the biocompatibility and imaging potential, MRI contrast agents should also have adequate in vivo biokinetics to achieve effective and accurate diagnosis of diseases. After intravenous injection, the physicochemical properties of the IONs highly determine their biological fate. These properties include the size and surface charge of the IONs along with their surface coating materials [21,22]. Referring to the size, IONs larger than 100 nm accumulate mainly in the liver and spleen. The primary cause of the high accumulation of the IONs in these organs is a process known as opsonization. Specifically, following the intravenous administration of the IONs into the blood circulation, proteins found in the plasma are rapidly adsorbed onto the surface of the IONs. The phagocytic cells (i.e., macrophages) found in the liver and spleen tend to recognize and engulf opsonins, ultimately leading to the phagocytosis of the IONs and their subsequent degradation into Fe^2+^ cations by the lysosomes. On the contrary, IONs less than 10 nm undergo renal clearance, whereas micrometer-sized IONPs are taken up by the lungs at a lower percentage. IONs with a size that ranges between 10 nm and 100 nm typically appear to have a prolonged retention time in the blood circulation and consequently can achieve higher accumulation in the target tissue, a feature that is ideal for an effective MRI contrast agent. Concerning the surface charge, almost neutral (minimally charged) IONs indicate minimal interactions with opsonins, hence they remain in blood circulation for longer times compared to IONs bearing positive or negative surface charge [21]. As far as the surface coating materials are concerned, their use is essential because they endow the IONs with hydrophilicity, biocompatibility, colloidal stability, and stealth properties in order to improve their in vivo effectiveness. For example, PEG, chitosan, dextran, and many other polymers are commonly used for surface coating of IONs, which are developed for medical purposes [23,24]. Additionally, surface coating imparts suitable surface chemistry to IONs, enabling the bioconjugation of imaging agents (i.e., radionuclides or fluorescent molecules) and targeting moieties (i.e., monoclonal antibodies, proteins, peptides, aptamers, etc.), resulting in the formation of IONs with enhanced diagnostic capabilities.

The delivery of IONs in the diseased area is accomplished by two targeting mechanisms. According to the first targeting mechanism, IONs exploit the increased vascular permeability and passively enter the fenestrations of the vessels. In addition, tumor tissues lack lymphatic drainage, thus preventing the IONs from escaping after they have entered into the tumor area. This behavior of passive targeting is called enhanced permeability and retention (EPR) effect. The mechanism of passive targeting is highly affected by the physicochemical properties of the IONs, the extent of the vascular rupture, and the draining rate of the lymphatic system. The second mechanism is referred to as selective (or active) targeting, since targeting moieties are attached onto the surface of the IONs to selectively target specific cells or tissues. The mechanism of selective targeting is affected by the physicochemical properties of the IONs and the number of the targeting moieties attached on their surface.

## 3. Diagnostic Radionuclides for SPECT and PET Imaging

In nuclear medicine, the diagnostic radiopharmaceuticals used for SPECT or PET imaging consist of a biologically active biomolecule (i.e., monoclonal antibodies, peptides, lipids, etc.) or small molecule (i.e., pharmacophore), which is radiolabeled with a photon (γ), positron (β^+^), or Auger emitting radionuclide. After the administration of the radiopharmaceutical in the patient’s body, it is accumulated to the diseased tissue (i.e., tumor) so as to highlight the biochemical changes related to the disease in question.

Table 1 indicates a list of radionuclides used for SPECT imaging. Due to its optimal physical properties, the metastable technetium-99 m (^99m^Tc) is the most frequently employed radionuclide, composing more than 80% of the diagnostic radiopharmaceuticals currently used in nuclear imaging. First of all, ^99m^Tc based radiopharmaceuticals can be easily synthesized and distributed to hospitals thanks to the half-life of 6 h, which also ensures that the patients receive the least dose of radiation possible. Secondly, the abundant (~89%) emission of γ-photons of 140 keV is preferred for high-resolution imaging. Thirdly, its ideal chemistry enables the radiolabeling with several chemical compounds. Finally, ^99m^Tc is daily provided by the in situ elution of a ^99^Mo/^99m^Tc generator [25,26]. The ^99m^Tc-based radiopharmaceuticals play an important role in oncology, as they are employed to detect malignancies or metastatic tumors. For example, ^99m^Tc-methylene-diphosphonate (^99m^Tc-MDP) or ^99m^Tc-hydroxymethane diphosphonate (^99m^Tc-HDP) are used in bone scintigraphy to diagnose cancer of the bones or metastases, and ^99m^Tc-nanocolloid is employed for imaging of the prostate’s sentinel lymph nodes. In addition to ^99m^Tc, other commonly used γ-emitting radionuclides are indium-111 (^111^In), iodine-123 (^123^I), iodine-125 (^125^I), and gallium-67 (^67^Ga) [26]. Radioiodine (^123^I and ^131^I) is used in the diagnosis of thyroid cancer, ^67^Ga for imaging of lymphoma, while ^111^In has been used to radiolabel peptides for the diagnosis of neuroendocrine tumors. Furthermore, lutetium-177 (^177^Lu) is a suitable radionuclide not only for SPECT imaging, but also for therapeutic applications, owing to its β^−^ emission of low energy (0.49 Mev) and γ emission (208 (11%), 113 (6.6%) keV), which can be imaged [26].

Table 2 indicates a list of radionuclides used for PET imaging. Such a radionuclide emits β^+^ (positron), which interacts with a free β^−^ (electron) in the tissue, releasing energy in the form of two oppositely emitted γ-photons of 511 keV that can be detected coincidently by PET camera. This process is known as annihilation.

Fluorine-18 (^18^F) is the most frequently used PET radionuclide due to its optimal physicochemical properties, including the abundant (~97%) emission of β^+^ of 511 keV, the low percentage of β^+^ of 0.640 MeV, and its ideal chemistry for radiolabeling with various molecular compounds [29]. In oncology, ^18^F is used in the form of fluorodeoxyglucose-18 (^18^F-FDG), a radiopharmaceutical used to diagnose cases of augmented consumption of glucose, which is usually associated with the appearance of malignancies. However, because of its short half-life of 1.83 h, other radionuclides, including copper-64 (^64^Cu) and zirconium-89 (^89^Zr), are also carefully studied. Among them, gallium-68 (^68^Ga) has received considerable attention for PET imaging. Firstly, owing to its half-life of 68 min, it assures that the patients are exposed minimally to radiation after the examination. Secondly, it forms stable conjugates with various chemical compounds due to its optimal chemical properties. Finally, its supply from a ^68^Ge/^68^Ga generator makes it readily available in every day clinical practice [30].

The IONs can be radiolabeled with various radiolabeling approaches, including the encapsulation of the radionuclide inside the core of iron oxide during the formation of the nanoparticles, the direct radiolabeling approach through which the radionuclide is conjugated on the surface of the IONs, which have been previously synthesized, and the chelator mediated approach through which either the radionuclide is conjugated to the complex chelator-IONs or the IONs are conjugated to the radiolabeled chelator, which has been previously synthesized [31,32]. DOTA (1,4,7,10-tetraazacyclododecane-1,4,7,10-tetraacetic acid), NOTA (1,4,7-tritazacyclononane-1,4,7-triacetic acid), and DTPA (diethylenetriaminepentaacetic acid) are common chelators, which are employed for the radiolabeling of the nanoparticles [32]. In addition, there is the proton/neutron beam activation method, which relies on the bombardment of the nanoparticles with a neutron or proton. In this case, an atom of the nanoparticle undergoes the nuclear reaction forming in situ the radionuclide. This radiolabeling method is characterized by the production of highly stable radiolabeled nanoparticles; however, it requires complex instrumentation for its application, and the high energies used in these nuclear reactions may affect the biological compounds that are attached onto the nanoparticle’s surface [32].

## 4. Passive SPECT/MRI DMCA

### 4.1. ^99m^Tc-Based Passive SPECT/MRI DMCA

The sentinel lymph node (SLN) is the first lymph node in which a tumor causes lymphatic drainage and the one that is more prone to the proliferation of metastatic cells. Therefore, a proper imaging and characterization of SLN are important to interpret the stage of cancer disease. Thus, Madru et al. [33] has developed a SPECT/MRI DMCA to achieve SLN imaging. The synthesis is carried out by direct radiolabeling polyethylene glycol (PEG) coated IONs with ^99m^Tc, using stannous chloride as the reducing agent. The RIONs are found to be highly stable for up to 24 h of incubation in both sterile water and human serum (namely, the intact percentage of the RIONs in sterile water is 98%, and in human serum, 97%). Following subcutaneous injection in Wistar rats, the RIONs exhibit a considerable uptake in the SLNs (100% injected dose per gram (ID/g)), while the uptake in the rest of the organs is negligible (<2% ID/g). The uptake of RIONs in the lymph nodes is also evident by SPECT and MR imaging, thus proving that the RIONs could be a promising DMCA for the imaging of breast cancer and malignant melanoma.

De Rosales et al. [34] have synthesized a ^99m^Tc based DMCA for SPECT and MR imaging. To develop the DMCA, the authors have directly conjugated Endorem, which is a clinically-approved ION for MR imaging of the RES system (namely, liver and spleen) with ^99m^Tc-dipicolylamine-alendronate (^99m^Tc-DTPA-ale), a bisphosphonate (BP) agent used for SPECT imaging. As indicated in Figure 1, the biodistribution of ^99m^Tc-DTPA-ale-Endorem (Figure 1A) is significantly different from the one of the free ^99m^Tc-DTPA-ale (Figure 1B), proving the high stability of BP-Endorem in vivo. The biodistribution data clearly shows that the ^99m^Tc-DTPA-ale-Endorem accumulates mainly in the liver and spleen, as expected for nanoparticles of this size (106 ± 60 nm). MR imaging (Figure 2A,B) is in good agreement with the biodistribution studies, showing significant signal loss in the liver and spleen where the^99m^Tc-DTPA-ale-Endorem has accumulated. The MRI data have also been confirmed by SPECT imaging (Figure 2C), thus evidencing the efficiency of dual imaging.

In the same context, Sandiford et al. [35] have synthesized ultrasmall RIONs for efficient SPECT/MRI imaging of blood vessels and vascular organs. The RIONs consist of IONs that are surface functionalized with a PEG polymer containing a bisphosphonate (BP) anchor and radiolabeled with ^99m^Tc. The radiolabeling of PEG-BP-IONs with ^99m^Tc is achieved by adding the ^99m^Tc-dipicolylamine-alendronate (^99m^Tc-DPA-ale), a bifunctional BP which has been reported [34,36] to radiolabel nanoparticles, exhibiting high radiolabeling stability not only in vitro, but also in vivo. According to the MRI findings, the non-radiolabeled PEG-BP-IONs are able to induce significant contrast enhancement in T_1_-weighted images, enabling a high-resolution visualization of the cardiovascular system. Additionally, SPECT imaging indicates the prolonged blood circulation times of the RIONs by providing hyperintense signals in the blood vessels and vascular organs, even at 200 min p.i.

Mirković et al. [37] have used ^99m^Tc radiolabeled IONs to investigate their in vivo biokinetics in healthy Wistar rats. The IONs are coated with a hydrophilic bisphosphonate ligand, which provides the appropriate surface chemistry for radiolabeling. The radiolabeling of bisphosphonate coated IONs with ^99m^Tc has been performed using stannous chloride as the reducing agent and has afforded a high radiolabeling yield of more than 95%. The RIONs also demonstrate excellent in vitro stability after 24 h of incubation in human serum. The in vivo evaluation of the RIONs in Wistar rats shows that they accumulate mainly in the RES organs, namely in the liver and spleen, at all examined time points (1, 2, and 24 h) after intravenous injection. This result is related to the fact that the RIONs are readily opsonized immediately after they enter into the blood circulation, which ultimately leads to their recognition and subsequent elimination by the macrophages (i.e., Kupffer cells). Scintigraphy studies have also been performed in Wistar rats immediately after the injection of RIONs. The results are in good agreement with the biodistribution data, demonstrating that these RIONs could serve as potential SPECT/MRI DMCA for diagnostic purposes.

For the detection and characterization of lesions within the body, Lee et al. [38] have developed an MRI contrast agent in a core-shell (CS) structure with superparamagnetic IONs in its core. The CS-IONs are also radiolabeled with ^99m^Tc in order to track their biodistribution with SPECT/CT imaging. As shown in Figure 3a, following the intravenous injection of the ^99m^Tc-CS-IONs in rats, a high radioactive signal is observed in the liver. The lungs, spleen, and bladder show significant accumulation of ^99m^Tc-CS-IONs, while negligible uptake is found in the kidneys and heart. ^99m^Tc-CS-IONs are 275 nm in size, which allows them to pass through the capillaries in the lungs, kidneys, and heart, but also, it is the main factor that contributes to their fast sequestration by the liver’s Kupffer cells. Figure 3b demonstrates coronal and axial T_2_ MR images of rat livers before and at 2 h, 2 weeks, and 4 weeks after the injection of CS-IONs. As it is indicated, the liver appears dark on T_2_ MR images as soon as the CS-IONs accumulate in the liver cells (2 h after administration). This is because the presence of the IONs changes the T_2_ relaxation time of the liver, leading to dramatic MR signal reduction. However, the observed signal reduction in the liver decreases with time due to the CS-IONs’ degradation by the lysosomes after their cellular internalization. The degradation of CS-IONs is also confirmed by histochemical methods using Prussian Blue staining for CS-IONs (Figure 3c). Indeed, the CS-IONs are observed at 2 h after injection, while at 2 weeks, the degradation of CS-IONs has diminished the Prussian blue staining.

Fu et al. [39] have evaluated the in vivo biodistribution pattern of ^99m^Tc based RIONs in normal rats at various time points up to 180 min p.i. Initially, the synthesis of the RIONs has been performed by the direct radiolabeling method, demonstrating a radiolabeling yield of more than 99%. In vitro stability studies show that the labeling yield is retained for a long period (i.e., 90% of the RIONs are intact according to instant thin layer chromatography), hence paving the way for further in vivo evaluation in animal models. The biodistribution studies performed in Wistar rats indicate that more than 80% of the injected RIONs are found in the liver in less than 30 min after injection, whereas a considerable accumulation is also found in the lungs, kidneys, and spleen, however at a lower extent compared to the accumulation in the liver. This result is attributed to the physicochemical characteristics of the RIONs, which highly affect their in vivo fate. Moreover, gamma-camera imaging is also performed in the rats after they have been intravenously injected with the RIONs and further confirms their accumulation in the area of the liver and lungs. Thus, based on the in vitro and in vivo results of the study, the authors have concluded that the RIONs show great potential for use in diagnostic applications.

Karageorgou et al. [40] have focused on the synthesis and subsequent in vitro and in vivo evaluation of RIONs for dual imaging with SPECT and MRI. The RIONs consist of ^99m^Tc radiolabeled and 2,3-dicarboxypropane-1,1-diphosphonic acid coated IONs (Figure 4a). Referring to the in vitro evaluation, the RIONs have been incubated in PBS and human serum for 24 h, exhibiting in vitro stability of 92.3% and 67.3%, respectively. For both media (PBS and human serum), the stability results are determined by thin layer chromatographic analysis, using acetone and citrate solution (0.1 M) as the mobile phases. In the case of the stability of RIONs in human serum, the acetone-developed chromatograms show no release of free ^99m^Tc from the RIONs incubated in human serum for up to 24 h [40]. According to the authors, the slight decrease observed (67.3%) is expected due to the presence of serum proteins that may bind ^99m^Tc, ultimately affecting the original DMCA [40]. As far as the in vivo evaluation is concerned, both biodistribution (Figure 4b) and imaging studies (namely, MRI (Figure 4c–h) and gamma-camera (Figure 4i)) have been performed in normal Swiss mice, showing consistent results by demonstrating noticeable liver uptake of the RIONs. Additionally, the low uptake observed in the stomach excludes the in vivo decomposition of the RIONs and the release of free ^99m^Tc. Most importantly, the MRI data also proves that the contrast efficacy of the RIONs (Figure 4e,h) is comparable to that of the non-radiolabeled IONs (Figure 4d,g).

### 4.2. ^111^In-Based Passive SPECT/MRI DMCA

Mousavie Anijdan et al. [41] have studied the biodistribution of indium-111 (^111^In) labeled dextran coated IONs in normal mice, which shows significant liver and spleen uptake (indicatively 69.8% for the liver and 21.4% ID/g for the spleen at 30 min p.i.), as well as fast clearance from the bloodstream (just 1% ID/g at 30 min p.i.). According to the results, the authors have suggested that the specific RIONs would serve as suitable SPECT/MRI DMCA for RES diagnosis.

### 4.3. ^125^I-Based Passive SPECT/MRI DMCA

Tang et al. [42] have used iodine-125 (^125^I) based RIONs to survey in vivo, through MR and SPECT imaging, mesenchymal stem cells transplanted intracerebrally or intravenously in rats suffering from ischemic stroke. Both MR and SPECT report that at 14 days after transplantation, the intravenously-implanted cells cannot be detected in the ischemic brain, and 90% of them are trapped in the lungs, while 35% of intracerebrally-implanted cells have migrated to the ischemic region.

### 4.4. ^177^Lu-Based Passive SPECT/MRI DMCA

Shanehsazzadeh et al. [43] have synthesized and evaluated the in vivo biodistribution of ^177^Lu radiolabeled IONs for potential use as a SPECT/MRI DMCA. The radiosynthesis of the DMCA is achieved through the use of the diethylenetriamene pentaacetate (DTPA) chelator, providing 99% radiochemical purity. The in vitro stability of the DMCA is found to be adequate, since ~80% of the RIONs have been found intact after their incubation for 48 h in human serum. The in vivo biodistribution results in normal rats demonstrate significant liver and spleen uptake at all time points studied, as well as rapid elimination from the blood circulation (just 1% ID/g at 30 min p.i.). According to the results, the authors have suggested that the specific DMCA would be appropriate for RES imaging, taking into account its high accumulation in the liver and spleen and its fast clearance from the other organs, mainly blood.

## 5. Selective SPECT/MRI DMCA

### 5.1. ^99m^Tc-Based Selective SPECT/MRI DMCA

Zhao et al. [44] have synthesized ^99m^Tc-radiolabeled and bevacizumab-conjugated ultrasmall IONs for active SPECT/MR imaging of hepatocellular carcinoma. The bevacizumab-conjugated IONs have been radiolabeled via the use of the bifunctional chelator DTPA, and afford high radiolabeling efficiencies of up to 92.9%. The in vivo SPECT/CT imaging performed in mice bearing hepatocellular carcinoma, HepG2, shows that the uptake of the DMCA in the tumor site exhibits an increased pattern from 1.29% (at 2 h p.i.) to 3.88% (at 24 h p.i.). The histological studies reveal the presence of iron in the HepG2 cancer cells, which further verifies the intratumoral uptake of the DMCA previously demonstrated by SPECT/CT.

Additionally, Tsiapa et al. [45] have developed and assessed ^99m^Tc labeled IONs conjugated with a new RGD derivate (cRGDfK-Orn3-CGG) to serve as selective SPECT/MRI DMCA for imaging of α_ν_β_3_-overexpressing tumor, namely U87MG glioblastoma. The selective DMCA has been assessed in vivo through biodistribution studies in healthy and U87MG glioblastoma tumor-bearing mice. The results demonstrate a significant uptake in the RES organs for both mice models. Referring to tumor uptake, the DMCA exhibits 9-fold higher tumor accumulation at 1 h p.i. (9.01 ± 0.19% IA/gr) compared to the non-selective one (1.05 ± 0.31% IA/gr). This result is also confirmed through the conduction of blocking experiments, indicating the specificity of the selective DMCA for α_ν_β_3_-integrin receptors compared to the non-selective one, whose tumor uptake is mainly accomplished through passive targeting.

Shanehsazzadeh et al. [46] have developed a selective DMCA for the SPECT/MR imaging of breast cancer. The DMCA consists of IONs, which are surface coated with dextran and radiolabeled with ^99m^Tc for SPECT imaging. The DMCA is also conjugated with the monoclonal antibody (mAb) C595 mAb to detect in vivo MUC1 antigen overexpressed in breast carcinomas. The in vivo evaluation of the DMCA has been conducted through biodistribution studies in MCU1 tumor-bearing BALB/c mice. Referring to the biodistribution data, the RES organs show the maximum uptake at 15 min p.i. (60% IA/g for the liver and 20% IA/g for the spleen), whereas the rest of the organs demonstrate minimal accumulation of the DMCA. However, at 24 h p.i. the accumulation in the RES organs is significantly reduced for both organs, reaching 24% IA/g in the liver and 5% IA/g in the spleen. As it has been mentioned previously, nanoparticles with sizes more than 100 nm undergo rapid opsonization upon entering the bloodstream. In this specific case, according to DLS measurements, the DMCA has a hydrodynamic size of 115 nm, which is sufficient enough to provoke the adsorption of opsonins on the DMCA, ultimately leading to its accumulation in the liver and spleen. However, the tumor to blood and tumor to muscle ratios have been found to be >10 and >55, respectively, despite the increased uptake in these organs.

De Souza Albernaz et al. [47] have synthesized and evaluated ^99m^Tc labeled IONs as potential selective DMCA for SPECT and MR imaging of breast cancer. For this reason, the nanoparticles have been conjugated with the human monoclonal antibody trastuzumab, which has the propensity to target HER2 receptors overexpressed in breast cancer cells. DTPA has also been used as a chelator for the subsequent radiolabeling with ^99m^Tc radionuclide. Biodistribution studies performed in both normal and BT-474 tumor-bearing mice have indicated that more than 80% of the injected DMCA undergoes renal clearance, while the uptake of DMCA in the liver is significantly low. This result shows that the DMCA is almost invisible from the immune system, allowing for its circulation in the blood for longer periods and ultimately facilitating its accumulation in the tumor. In fact, more than 20% of the injected dose is detected in the tumor site, verifying the use of the DMCA in cancer diagnosis. 

Tsoukalas et al. [48] have developed a selective DMCA for SPECT/MR imaging of tumor vascularization. In particular, the DMCA consists of IONs, which have been surface-coated with dimercaptosuccinic acid (DMSA) and radiolabeled with ^99m^Tc. The presence of DMSA endows the surface of the RIONs with the appropriate functional groups for further functionalization with the monoclonal antibody bevacizumab (BCZM), thus forming the selective DMCA. According to the results, the binding efficiency of the antibody-conjugated DMCA is successfully demonstrated on M-165 tumor cells by targeting the VEGF-165 isoform. The selective character of the DMCA is further confirmed by the biodistribution studies, which indicates increased tumor accumulation up to 4 h p.i. (~17% ID/g), followed by a slight decrease at 24 h p.i. (~7% ID/g). At 24 h p.i., the ratios of tumor to blood and tumor to muscles exhibit their maximum values, namely 7 and 18, respectively. Except for the tumor, the biodistribution pattern of the selective DMCA shows significantly lower liver uptake (~18% ID/g) compared to the non-selective one (~25% ID/g). In the spleen, the accumulation is considerably lower for both selective and non-selective DMCAs (9% ID/g and 5% ID/g at 24 h p.i., respectively). The presence of the antibody has a significant impact on the biodistribution of the selective DMCA, as seen from the low RES uptake. Additionally, high uptake of this DMCA is found in the kidneys and lungs. The imaging results, acquired from gamma camera and MRI, are found to be consistent with the biodistribution ones, thus demonstrating the efficacy of the selective DMCA for targeted imaging of tumor vascularization.

### 5.2. ^111^In-Based Selective SPECT/MRI DMCA

A selective DMCA has been developed by Misri et al. [49] for SPECT and MR imaging of mesothelin-expressing cancers. For this purpose, the selective DMCA has been synthesized by conjugating an ^111^In radiolabeled antimesothelin antibody (mAbMB) to carboxy-methyl dextran coated IONs, ultimately forming the ^111^In-mAbMB conjugated DMCA. The radiolabeling of the mAbMB with ^111^In is achieved with the use of a DTPA chelator. The uptake of the selective DMCA is significantly higher in the mesothelin positive cancer cell line (A431K5) compared to the mesothelin negative one (A431), as it is demonstrated in vitro through cell binding studies. In vivo biodistribution studies also evidence the DMCA accumulation in the A431K5 tumor, which remains up to 4.8% ID/gr at 72 h p.i. However, the ^111^In-mAbMB conjugated DMCA also shows a high accumulation in the RES organs (48.38% ID/gr for the spleen and 8.14% ID/gr for the liver at 72 h p.i.), despite its selective character. The biodistribution results are enhanced by the MR imaging, which also indicates the uptake of the ^111^In-mAbMB conjugated DMCA in the A431K5 tumor. According to their findings, the authors have suggested that the ^111^In-mAbMB conjugated DMCA has the potential to be employed in future SPECT/MR imaging and treatment of cancer.

Zolata et al. [50] have used IONs to synthesize a selective SPECT/MRI DMCA for the imaging of HER2 overexpression in breast cancer. In particular, the IONs have been functionalized with N-Hydroxysuccinimide (NHS) ester of Polyethylene Glycol Maleimide (NHS-PEG-Mal) to enable conjugation with thiolated 3,6,9,15-tetraazabicyclo[9.3.1]pentadeca-1(15),11,13-triene-3,6,9,-triacetic acid (PCTA) bifunctional chelator for radiolabeling with ^111^In and with Trastuzumab antibody for targeting HER2 overexpressing tumors. In vivo standard biodistribution in breast tumor-bearing BALB/c mice exhibits significantly high accumulation of the selective DMCA in the tumor, while non-specific uptake is observed in mice with blocked HER2 receptors. In vivo SPECT and MRI data are found to be consistent with the ones obtained from the biodistribution studies, proving the dual imaging capability of the selective DMCA.

### 5.3. ^125^I-Based Selective SPECT/MRI DMCA

Wang et al. [51] have synthesized a selective DMCA, using polyethylene glycol coated IONs conjugated with a c(RGDyK) peptide and radiolabeled with ^125^I, for in vivo SPECT/MR imaging of U87MG glioblastoma. The radiolabeling yield of the RIONs, namely ^125^I-c(RGDyK)-DMCA, is determined to be ~60% by the radiochromatography analysis, while after purification the radiochemical purity of ^125^I-c(RGDyK)-DMCA is higher than 99%. The in vitro stability of ^125^I-c(RGDyK)-DMCA has been tested in saline solution for 14 days and is found to be highly stable (more than 88% intact). In vivo biodistribution studies performed in U87MG tumor-bearing mice indicate an increased accumulation of the ^125^I-c(RGDyK)-DMCA in the tumor, which peaks at 6 h p.i. (~7% ID/g) (Figure 5a). This result is associated to the strong binding affinity of the ^125^I-c(RGDyK)-DMCA for the tumors expressing integrin α_ν_β_3_. Then, the tumor uptake decreases to 0.1%, indicating that the DMCA gradually re-enters into the bloodstream. The accumulation in the liver and spleen is found to be 8.5% and 1.9% respectively, at 1 h p.i., which thereafter gradually decreases for both organs (<1.5% at 48 h p.i.) (Figure 5a). The high in vivo U87MG glioblastoma targeting efficacy and low RES uptake of the DMCA is further confirmed by SPECT and MR imaging. As demonstrated in Figure 5b, the signal produced by ^125^I-c(RGDyK)-DMCA in the tumor site gradually increases up to 12 h p.i., showing a maximum uptake at 6 h p.i. Afterwards, the signal decreases and is undetectable. The radioactive signal in the tumor of the blocking group (mice administered with free RGD and ^125^I-c(RGDyK)-DMCA) is undetectable at all time points. Consistent results have also been observed with MRI (Figure 5c).

Deng et al. [52] have evaluated a potential SPECT and MRI DMCA for in vivo dual imaging of integrin α_ν_β_3_ expression in breast cancer. The DMCA consists of IONs, which are radiolabeled with ^125^I and surface functionalized with the peptide arginine-glycine-aspartic (cRGD). Owing to the presence of the cRGD, the ^125^I-cRGD DMCA shows a substantial accumulation in the tumor, peaking at 8% at 4 h p.i. This result is due to the fact that the DMCA has a small hydrodynamic size, which renders it partly invisible to the immune system. Indeed, at 4 h p.i. the accumulation of the ^125^I-cRGD DMCA in the liver is found to be lower than 10%, while at the same time the presence of the DMCA in the blood is higher than 30%. The delivery of the ^125^I-cRGD DMCA in the tumor site is also demonstrated by in vivo SPECT/MR imaging studies, thus suggesting its application in the early diagnosis of cancer.

Liu et al. [53] have successfully developed RIONs to serve as a novel DMCA for in vivo detection of tumors by both MRI and SPECT. The RIONs are composed of biocompatible Fe_3_O_4_ nanocrystals, an antigastric cancer monoclonal antibody, 3H11, and ^125^I radionuclide. Particularly, the antibody is covalently bound onto the surface of the nanocrystals through the carboxylic groups of their surface coating, namely α,ω-dicarboxyl-terminated PEG (HOOC-PEG-COOH). The antibody is also radiolabeled with ^125^I for SPECT imaging. Then, a series of in vivo experiments are carried out to detect BGC823 human gastric cancer xenografts in BALB/c nude mice by MRI and gamma-imaging modalities. The biodistribution of the ^125^I-3H11 RIONs has also been investigated in nude mice bearing the BGC823 tumor. The results of the biodistribution study demonstrate high accumulation of ^125^I-3H11 RIONs in the liver (38.4 ± 3.6% ID/organ) within 10 min p.i. and increased blood retention (51.2 ± 6.8% ID/organ). From then on, the liver uptake starts to decrease, while the uptake in the tumor site increases within 24 h p.i. to 4.1 ± 0.6% ID/organ and reaches a maximum 48 h p.i. (4.7 ± 0.7% ID/organ). Eventually the RIONs are accumulated more in the tumor than in the liver. The RIONs also exhibit an excellent circulation behavior in the blood remaining in the bloodstream up to the third day p.i. The biodistribution pattern of ^125^I-3H11 RIONs is in consistence with the one obtained by MRI and gamma imaging studies, hence proving that these RIONs could constitute a powerful tool for selective tumor imaging.

## 6. Passive PET/MRI DMCA

### 6.1. ^64^Cu-Based Passive PET/MRI DMCA

A novel PET/MRI DMCA, based on superparamagnetic IONs, has been developed by Glaus et al. [54]. The iron oxide core of the nanoparticles is coated with PEGylated phospholipids, on the termini of which the chelator, DOTA, is conjugated for radiolabeling with copper-64 (^64^Cu). According to the study, the RIONs indicate excellent in vivo stability in mouse serum over the course of 24 h incubation, thus producing high PET and MR signals. The potential of the RIONs has been investigated through biodistribution studies and PET/CT imaging. The findings of both studies correlate well, indicating high initial blood retention (47.5 ± 4.6% ID/g at 10 min p.i. and 37.3 ± 12.9% ID/g at 1 h p.i.) and moderate RES uptake (15.9 ± 1.4% ID/g at 10 min p.i., 33.4 ± 1.9% ID/g at 1 h p.i. for the liver and 10.1 ± 0.6% ID/g at 10 min p.i., 19.9 ± 2.3% ID/g at 1 h p.i. for the spleen), thus rendering the RIONs as attractive PET/MRI DMCA for future application in disease studies.

Xie et al. [55] have utilized dopamine to modify the surface of IONs, resulting in nanoconjugates that can be easily encapsulated into human serum albumin (HSA) matrices. Then, the HSA-IONs are radiolabeled with ^64^Cu via the use of a DOTA chelator to achieve PET imaging. The HSA-IONs are further labeled with Cy5.5 to perform near-infrared fluorescence (NIRF) imaging as well, and are subsequently evaluated in a mouse with U87MG tumor. The behavior of ^64^Cu-HAS-IONs is carefully investigated with in vivo PET/NIRF/MR imaging, ex vivo analyses, and histological examinations. The overall findings indicate that ^64^Cu-HAS-IONs exhibits prolonged circulation time in the blood, along with high rates of extravasation and retention at the tumor. Such ideal characteristics are mainly related to the compact HAS coating that endows the IONs with stealth properties, enabling them to circulate in the bloodstream for longer periods, ultimately reaching the tumor site. This is why, according to the histological examinations, the accumulation of ^64^Cu-HAS-IONs in the tumor is not attributed to macrophage uptake. The authors have concluded that these observations are encouraging, considering that the ultimate goal of the utilization of ^64^Cu-HAS-IONs is to serve as DMCAs in diagnostic applications.

### 6.2. ^68^Ga-Based Passive PET/MRI DMCA

Burke et al. [56,57] have referred to the development and subsequent radiolabeling with ^68^Ga of iron oxide nanorods in order to be used in the detection of liver malignancies with PET and MRI. The radiolabeling with ^68^Ga has been accomplished via the direct method, without the use of any chelator, resulting in the formation of highly stable conjugates. This result is attributed to the fact that the nanorods have been surface functionalized with silica coating, which provides the nanorods with the appropriate surface chemistry to form strong bonds with ^68^Ga. Subsequently, the biodistribution of the radiolabeled nanorods is evaluated with PET and MR imaging. Both studies evidence the high accumulation of the radiolabeled nanorods in the liver of CD1 female nude mice within 5 min after injection, hence dictating its application for dual-imaging of liver malignancies.

Karageorgou et al. [10,58] have synthesized ^68^Ga-based RIONs and subsequently have investigated their in vivo biodistribution and imaging efficiency with PET and MRI. After radiosynthesis, the RIONs exhibit a satisfactory radiolabeling yield (~70%), while after purification they show a radiochemical purity of approximately 91%. The in vitro stability of RIONs has been examined up to 2 h after incubation with human serum and has been found to be highly stable (more than 92% intact). The biodistribution results of the intravenously injected ^68^Ga-based RIONs in normal Swiss mice show that the RIONs are mainly found in the RES organs, where they remain up to 120 min p.i. (Figure 6a). Referring to the in vivo imaging studies performed in normal Swiss mice, both PET (Figure 6b–d) and MRI (Figure 6e–g) show identical results by demonstrating the high uptake of the RIONs in the area of the liver and spleen. Additionally, comparing the data of Figure 6e,f, it is clearly demonstrated that the contrast efficacy of the RIONs is concentration dependent. Furthermore, the MRI data prove that the contrast efficacy of the RIONs (Figure 6f) is similar to that of the non-radiolabeled IONs (Figure 6g) at a three-times higher concentration.

The biodistribution pattern of ^68^Ga based RIONs has been evaluated in mice by Lahooti et al. [59]. The RIONs consist of IONs, which are surface coated with PEG polymer and radiolabeled with ^68^Ga, ultimately forming the ^68^Ga-PEG-IONs. According to the findings of the study, high accumulation of the RIONs is observed in the liver less than 5 min after their administration into the bloodstream (37.41% IA/g at), which further increases at 120 min p.i. (60.62% IA/g). The same holds for the spleen, where the RIONs uptake increases from 9.17% IA/g at 5 min p.i. to 12.65% IA/g at 120 min p.i. The authors have concluded that both the hydrodynamic size of the RIONs, namely 85 nm as it was determined by DLS, and the coating material used for surface functionalization, namely PEG polymer, highly affect their in vivo fate.

S. Papadopoulou et al. [60] have radiolabeled IONs with ^68^Ga to serve as potential DMCA in medical imaging. To achieve this goal, the IONs are coated with p(MAA-g-EGMA), a copolymer which endows the IONs with the necessary functional groups for further functionalization. For radiolabeling, the authors have used both chelator-mediated and the chelator-free approaches. Referring to the chelator-mediated radiolabeling approach, the authors have conjugated the chelator NODAGA-NHS onto the surface of the nanoparticles for the subsequent complexation with ^68^Ga, ultimately forming the ^68^Ga-Mag-NODAGA DMCA. Referring to the chelator-free radiolabeling approach, ^68^Ga has been directly conjugated onto the surface of the nanoparticles, ultimately forming the ^68^Ga-Mag DMCA. Both DMCAs exhibit high radiolabeling (>92%) and satisfactory in vitro stability efficiencies (>80% of both DMCAs remain intact after 2h of incubation with human serum). Then, the in vivo biodistribution of the DMCAs has been studied in normal Swiss and 4T1 tumor-bearing SCID mice, indicating that the DMCAs highly accumulate in the liver and spleen in both mice models. All other organs exhibit minor accumulation. Such an in vivo behavior is associated with the size and the negative surface charge of the DMCAs, which determine at a high extent their interactions with plasma proteins. According to DLS measurements, the size of ^68^Ga-Mag-NODAGA is found to be 108 nm, while that of ^68^Ga-Mag is 87 nm. Zeta potential measurements show a charge of −37 mV for both DMCAs. Moreover, the tumor uptake for both DMCAs follows an increasing trend, reaching at 2 h p.i. the 1.83 ± 0.27% ID/g for ^68^Ga-Mag-NODAGA and the 1.06 ± 0.35% ID/g for ^68^Ga-Mag. However, both DMCAs demonstrate tumor to muscle ratios well above unity. Subsequently, in vivo PET imaging studies performed in tumor-bearing mice are in direct agreement with the respective biodistribution results, thus indicating the potential of these DMCAs in diagnostic applications.

Almasi et al. [61] have developed and evaluated a DMCA based on ΙOΝs for PET and MR imaging of tumors. The synthesis of the DMCA has been performed by coating ΙOΝs with the dendrimer PAMAM, a three-dimensional and hyperbranched polymer that provides functional groups in its terminals for further functionalization. Indeed, PAMAM dendrimer is subsequently conjugated with the DOTA chelator and then radiolabeled with ^68^Ga radionuclide. The in vitro MRI measurements show that the DMCA could serve as an effective T_2_-weighted contrast agent, exhibiting an increased relaxation value of 96.7 mM^−1^s^−1^. Moreover, the DMCA is mainly taken up by the RES organs, as shown by the in vivo biodistribution studies. However, despite this observation, the tumor uptake is found to be satisfactory (3.4% ID/g). The results are further confirmed and clarified by PET imaging, indicating that the specific DMCA could be used in the diagnosis of tumors via PET and MR imaging.

Evertsson et al. [62] have synthesized ^68^Ga-based RIONs for in vivo pre-operative PET/MR imaging, followed by intra-operative localization of sentinel-lymph nodes in rats with magnetomotive ultrasound. In this particular case, the authors have successfully managed to combine the high sensitivity and quantification capabilities of PET and the high resolution of MRI together with the magnetomotive ultrasound for accurate detection of lymph nodes.

Cho et al. [63] have evaluated ^68^Ga based RIONs for future PET/MR imaging of cancer. According to the study, the RIONs show increased cellular uptake in colon (CT-26) and breast (SK-BR-3) cancer cells at 120 min after incubation (8.8% and 15.5% respectively), thus establishing the potential application of these RIONs in cancer diagnosis.

### 6.3. ^11^C-Based Passive PET/MRI DMCA

Sharma et al. [64] have synthesized and assessed the in vivo imaging effectiveness of a passive PET/MRI DMCA in animal models. To this effect, the authors have synthesized IONs with carboxyl (-COOH) and amine (-NH2) functional groups provided by the ligands on the nanoparticles’ surface. Both functional groups enable the direct radiolabeling of the IONs with the positron emitting radionuclide, carbon-11 (^11^C). In order to investigate the proof-of-concept dual-modality PET/MR imaging using the ^11^C based RIONs, initially T_2_-weighted MR images have been obtained before and after intravenous injection of non-radiolabeled IONs in Swiss Webster mice. The MRI data, presented in Figure 7, demonstrates a significant signal loss (or negative contrast enhancement) in the liver of the mouse after the injection of the non-radiolabeled IONs (Figure 7b) compared to the one before the injection (Figure 7a). Simultaneous PET and MR imaging has been conducted in a mouse at 15 min after injection with ^11^C based RIONs (Figure 8). Focusing on the liver of the mouse, both PET and MR images indicate the presence of the ^11^C based RIONs in this organ. The advantage of the synergistic effect of simultaneous PET/MR imaging is clear when comparing the images of the bottom line of Figure 8 to images of either top or middle lines acquired by each technique alone.

### 6.4. ^89^Zr-Based Passive PET/MRI DMCA

Unak et al. [65] have investigated zirconium-89 (^89^Zr) radiolabeled and TiO_2_ conjugated IONs to be used as a multifunctional agent for both diagnosis and therapy of cancer. Indeed, the presence of TiO_2_ can enable photodynamic therapy of cancer, while the IONs, along with the radionuclide, are useful for MR and PET imaging, respectively. The results of the study show that ^89^Zr-Fe_3_O-TiO_2_ nanoparticles are feasible as PET/MRI DMCA with potential for photodynamic therapy in prostate cancer cells.

Thorek et al. [66] have developed a PET/MRI DMCA for dual-imaging of the axillary and brachial lymph node drainage in mice, by radiolabeling clinically used carbohydrate-coated IONs, namely ferumoxytol, with the positron-emitter ^89^Zr, ultimately forming ^89^Zr-ferumoxytol. The radiolabeling of ferumoxytol with ^89^Zr has been achieved with the chelator, DFO, which is covalently attached onto the carboxymethyl coating of IONs. The RIONs, ^89^Zr-ferumoxytol, successfully detect lymph node drainage in preclinical mice models with high sensitivity and accuracy by means of PET/MR imaging.

### 6.5. ^18^F-Based Passive PET/MRI DMCA

Belderbos et al. [67] have evaluated the potential of fluorine-18 (^18^F) radiolabeled Fe_3_O_4_@Al(OH)_3_ nanoparticles as in vivo mesenchymal stem cell (mMSC) tracking agents for simultaneous PET/MRI in healthy mice. Specifically, the biodistribution of both ^18^F-Fe_3_O_4_@Al(OH)_3_ and mMSCs labeled with these RIONs has been studied in C57Bl/6 mice models using simultaneous PET/MR imaging. Both PET and MR imaging results evidence that the ^18^F-Fe_3_O_4_@Al(OH)_3_ RIONs are mainly accumulated in the liver, while the accumulation of radiolabeled mMSCs in the lungs is only provided with PET. Most importantly, the absence of any adverse side effects on blood cells after the injection of either RIONs or radiolabeled mMSCs proves their biocompatibility in vivo. Finally, the authors have concluded that the particular RIONs and ^18^F-Fe_3_O_4_@Al(OH)_3_ could be employed as promising candidates for simultaneous PET/MR cell tracking applications.

### 6.6. ^69^Ge-Based Passive PET/MRI DMCA

PEGylated superparamagnetic IONs have been synthesized by Chakravarty et al. [68] and were subsequently radiolabeled with germanium-69 (^69^Ge) to serve as potential PET/MRI DMCA for dual imaging of sentinel lymph nodes. Due to the high affinity of ^69^Ge for metal oxides, the radiolabeling has been achieved by incorporating ^69^Ge into the core of the nanoparticles via a chelator-free approach. The in vivo imaging efficacy of the DMCA has been investigated by PET and MRI studies in normal BALB/c mice. Referring to the PET results, they clearly demonstrate accumulation of the DMCA in the popliteal lymph node at 0.5, 2, and 20 h p.i. (7.5 ± 2.5, 12.5 ± 3.1 and 28.0 ± 5.2% ID/g, respectively). The accumulation of the DMCA in the popliteal lymph node could also be imaged by MRI, which shows significant darkening (i.e., signal loss) of the lymph node due to the presence of the DMCA, thus establishing this ^69^Ge-based DMCA as an important candidate for future simultaneous PET/MR imaging.

## 7. Selective PET/MRI DMCA

### 7.1. ^64^Cu-Based Selective PET/MRI DMCA

In a study performed by Lee et al. [69], RIONs have been synthesized for in vivo detection of tumors that overexpress integrin α_ν_β_3_. The RIONs consist of polyaspartic acid (PASP) coated IONs, which are also conjugated with RGD, to accomplish the selective targeting of integrin and DOTA, for radiolabeling with ^64^Cu, ultimately forming ^64^Cu-DOTA-IONs-RGD RIONs (Figure 9a). The targeting efficacy of the RIONs are investigated in vivo by dividing the tumor-bearing mice into three groups. The first group is injected with ^64^Cu-DOTA-IONs, the second group is injected with the ^64^Cu-DOTA-IONs-RGD along with RGD peptide for RGD blocking, and the third group is injected with the ^64^Cu-DOTA-IONs-RGD. PET imaging data, illustrated in Figure 9b, shows prominent RES accumulation of the RIONs, namely ^64^Cu-DOTA-IONs and ^64^Cu-DOTA-IONs-RGD, at all examined time points (1, 4, 21 h) for all mice groups. However, the U87MG tumor in the third group of mice presents high signal intensity at all time points compared to the other group of mice, and the maximum signal intensity is observed at 4 h p.i. Such a result confirms the specificity of RGD peptide for α_ν_β_3_ integrin positive tumors. The MRI data (Figure 9d–k) are in complete agreement with the PET results, showing a negative contrast enhancement in the tumor of mice injected with non-radiolabeled DOTA-IONs-RGD (Figure 9f,j) compared to the ones injected with either the non-radiolabeled DOTA-IONs (Figure 9e,i) or the DOTA-IONs-RGD along with the blocking agent (Figure 9g,k). Additionally, the strong signal loss in the area of RES organs compared to the control group (Figure 9d,h) is caused by the presence of both DOTA-IONs and DOTA-IONs-RGD.

Similarly, Yang et al. [70] have synthesized a PET/MRI DMCA based on IONs for selective targeting of tumors that overexpress integrin α_ν_β_3_. The DMCA has been surface functionalized with polyethylene glycol (PEG), which serves as an effective linker for conjugation with the targeting moiety, cRGD peptide, and with the macrocyclic chelator, 1,4,7-triazacyclononane-N, N′, N″-triacetic acid (NOTA), for subsequent radiolabeling with ^64^Cu. The specificity of the DMCA for tumors expressing α_ν_β_3_ integrin is evident in both PET and biodistribution studies, since they clearly indicate the increased uptake of the DMCA in the tumor compared to its free-cRGD counterpart. Moreover, in vitro relaxivity measurements demonstrate that the MR imaging potential of the DMCA is comparable with the one obtained from a conventional contrast agent frequently used in MR imaging.

Shi et al. [71] have synthesized and evaluated in vivo a novel PET/MRI DMCA based on 64Cu radiolabeled manganese ferrite nanoparticles. The DMCA is further conjugated with cyclic arginine-glycine-aspartic acid (RGD) peptide for in vivo targeting of the ανβ3 receptor in U87MG tumor-bearing mice. Briefly, the study reveals that the DMCA has a high r_2_ relaxivity value (267.5 mM^−^^1^s^−^^1^), a property which enhances its MR imaging effect, as well as adequate specificity to the ανβ3 receptor. The biodistribution of the DMCA in the tumor-bearing mice shows significant accumulation in the liver (25.04 ± 0.64% ID/g at 3 h p.i.), while lower uptake is found in the spleen and kidneys (7.38 ± 0.33% ID/g and 8.49 ± 0.76% ID/g at 3 h p.i., respectively). Referring to the tumor, the uptake of the DMCA is found to be 5.69 ± 0.56% ID/g, which is significantly higher compared to the one found for the RGD-free DMCA (be 2.70 ± 0.63% ID/g). The localization of the DMCA in the tumor site is also evident by both PET/CT and MR imaging modalities.

### 7.2. ^68^Ga-Based Selective PET/MRI DMCA

Moon et al. [72] have developed a ^68^Ga-based DMCA to specifically target prostate cancer. Particularly, IONs are radiolabeled with ^68^Ga via the use of the chelator, DOTA, which has been conjugated onto the surface of the nanoparticles in advance. Then, the DMCA is also conjugated with the glutamate-urea-lysine (GUL) to selectively target prostate specific membrane antigen (PSMA)-positive tumors (Figure 10a). The MR imaging of the DOTA-IONs-GUL contrast agent is performed in a BALB/c mouse with 22Rv1 and PC-3 tumors at its left and right flank, respectively. As shown in Figure 10c,d, the contrast agent is taken up by the (PSMA)-positive tumor, namely 22Rv1. Moreover, PET imaging not only confirms the specific accumulation in the 22Rv1 tumor, but also provides quantitative information (Figure 10b). Thus, the ^68^Ga-DOTA-IONs-GUL DMCA indicates promising dual-imaging potential for highlighting prostate cancer by exploiting the complementary advantages of each imaging techniques.

Similarly, Kim et al. [73] have designed a ^68^Ga-based DMCA to selectively target colon cancer by radiolabeling IONs with ^68^Ga and conjugating them with oleanolic acid, which serves as the targeting moiety. To achieve radiolabeling with ^68^Ga, the IONs are coupled with the macrocyclic chelator, NOTA. Due to the presence of the oleanolic acid, the selective DMCA is able specifically detect colon cancer cells (HT-29) both in vitro and in vivo using HT-29 xenograft mice models. Then, fusion PET/MRI images of the tumor are obtained, proving the effectiveness of the DMCA for future use in colon cancer diagnosis.

Gholipour et al. [74] have synthesized ^68^Ga labeled thiosemicarbazone functionalized IONs to selectively target 4T1 breast cancer. The radiolabeling yield of the newly formed RIONs is found to be more than 95% at 60 °C, also exhibiting excellent in vitro stability in cysteine and blood rerum, hence proving the strong binding of ^68^Ga in thiosemicarbazones. To achieve the specific targeting, the RIONs are also bioconjugated with biotin, as the 4T1 cancer cells overexpress positive receptors for this biomolecule. The in vivo evaluation of the selective RIONs is performed in 4T1 tumor and non-tumor bearing immunodeficient mice. Referring to the first mice model, the findings show that despite the observed significant RES uptake, 5.4% of the injected RIONs are found in the tumor, supporting their specificity in 4T1 tumors. According to in vitro characterization of the RIONs, they exhibit a negative charge of −15.1 mV and a hydrodynamic size of 114 nm, features that enhance the opsonization of the RIONs and their subsequent elimination from the immune system. Referring to the second mice model, the biodistribution study shows consistent results with the ones acquired from the tumor-bearing mice model. The tumor uptake is also evident by PET-CT imaging, thus evidencing the potential of the RIONs to be employed as PET/MRI DMCAs in the diagnosis of 4T1 breast cancer.

Wei et al. [75] have prepared a PET/MRI DMCA based on extremely small IONs to inspect the acidic tumor microenvironment. For this reason, the DMCA is conjugated with pH-low insertion peptides, which serve as the targeting moieties. The in vivo MRI study shows that the intravenously injected DMCA proves to be an efficient T_1_-weighted contrast agent, despite the fact that it demonstrates low tumor uptake. The in vivo PET imaging verifies the specific accumulation of the DMCA in the tumor area only after its intratumor injection. Although the tumor uptake of the intravenously injected DMCA is low enough to be detected with PET imaging, the overall results of the study indicate that the DMCA has all the necessary properties to be employed for PET/MR imaging of the acidic tumor microenvironment.

Hajiramezanali et al. [76] have synthesized and evaluated both in vitro and in vivo a potential selective PET/MRI DMCA for breast cancer detection. In particular, chitosan-coated IONs have been radiolabeled with ^68^Ga through the use of DOTA chelator and conjugated with the targeting peptide, bombesin. According to the in vitro results of the study, the selective DMCA exhibits ideal hydrodynamic size (between 20 and 30 nm), insignificant toxicity (>74% cell viability), excellent human serum stability (92% intact after 2 h of incubation), and a high binding affinity for gastrin-releasing peptide receptors overexpressed in breast tumors. Furthermore, the results of the biodistribution studies exhibit a rapid RES accumulation in both healthy and T-47D breast tumor-bearing mice. However, the tumor uptake shows an increased pattern over time and reaches up to 2.27% ID/g at 2 h p.i. Both PET and MR imaging studies evidence the accumulation of the DMCA in the T-47D tumor, thus enhancing its effectiveness as a potential imaging agent for breast cancer detection. Probably one of the last issues that should be discussed in a review article refers to the optimal choice of the chelating agent (DOTA-type or NOTA-type) in terms of thermodynamic and kinetic stability towards ^64^Cu^2+^ and ^68^Ga^3+^ labeling. Briefly, the chelator NOTA has a small ligand cavity and is selective to small metal ions, such as Ga^3+^ and Cu^2+^. The formed [Cu(NOTA)]^-^ complex is hexacoordinated at pH7, and kinetic studies have shown that complex formation is very rapid (milliseconds). The NOTA ligand forms thermodynamically stable complexes even at low pH (logK_ML_ = 23.33); however, the Cu(II) complex exhibits low kinetic inertness [77]. In the case of DOTA, the formed Cu(II) complexes exhibit good thermodynamic stability (logK_ML_ = 22.3) and adequate in vivo stability, but DOTA requires more time (~1 s) and heating for complex formation. All in all, NOTA appears to be the better ligand for complexing Cu^2+^. Gallium^3+^ is hexacoordinated to DOTA, resulting in an N_4_O_2_ distorted octahedron. It has been shown that the large size of the macrocycle is not ideal for Ga^3+^ complexation (logK_1_ = 26.05) [78]; however, Ga^3+^ complexes with DOTA exhibit high thermodynamic stability (logK_ML_ = 21.3) and have excellent in vivo stability [79]. Despite the limitations associated with the use of DOTA in biomolecule radiolabeling, to this day two of the four ^68^Ga-based radiopharmaceuticals in clinical practice are based on the DOTA chelator. Finally, another important issue refers to the comparison between passive and selective accumulation onto the tumor in both qualitative and quantitative terms. Upon administration in blood circulation, passive and selective DMCAs should possess specific physicochemical characteristics (briefly discussed in Section 2) for successful delivery to the tumor site of interest. These characteristics will endow them with the desired stealth properties to evade, as much as possible, the clearance by the phagocytic cells of the RES organs [21]. Both in passive and in selective targeting, the longer the DMCAs remain in the bloodstream, the higher likelihood they have to reach and accumulate into the tumor via the EPR effect. Ultimately, the accumulation and penetration of both passive and selective DMCAs is highly influenced by their size, shape, and surface charge [80]. Most of the studies recalled in this review [44,45,46,47,48,49,50,51,52,53,69,70,71,72,73,74,75] give evidence that the accumulation of selective DMCAs at the tumor is increased compared to passive ones. However, the tumor uptake is overshadowed by the RES uptake. In the same context, studies [81,82,83] conducted on passive and selective nanoparticles (i.e., gold, lipidic and siRNA) have reported that selective targeting neither increases the overall accumulation nor changes the biodistribution of these nanoparticles at the tumor. Notably, these selective nanoparticles have enhanced cellular uptake/internalization into the cancer cells in comparison to passive ones. Lastly, some studies [81,83,84] have demonstrated that selective nanoparticles are spatially distributed more homogeneously over the tumor, ultimately resulting in better imaging performance.

## 8. Conclusions

Due to their multifaceted physical properties, RIONs serve as the parent platform for the development of powerful DMCAs utilized in SPECT/MRI and PET/MRI applications. Their main advantage originates from the fact that they can be employed in the above mentioned complementary, minimally invasive imaging modalities simultaneously, ultimately providing the optimal combination of the highest possible sensitivity with the maximum achievable spatial resolution. We seek out that in the near future, research goals should focus on the production of RIONs with ideal characteristics that affect their in vivo fate and, consequently, their targeting and imaging potential. With the continuous development of new RIONs, the future of DMCAs is definitely promising in the clinical diagnosis of diseases.

## Figures and Tables

**Figure 1 nanomaterials-13-00503-f001:**
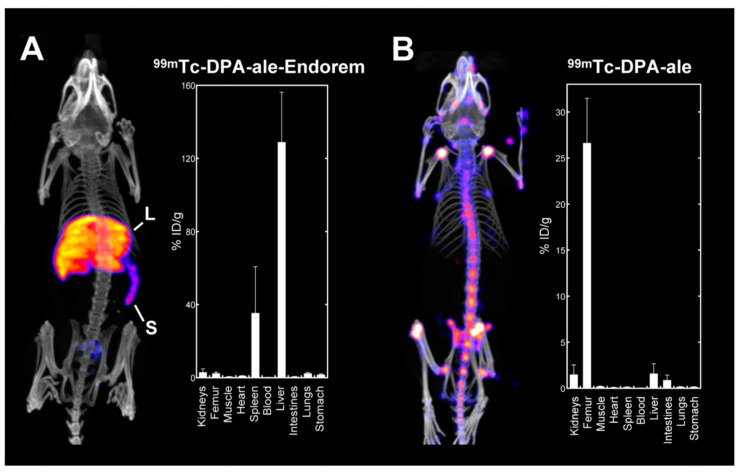
Whole body SPECT-CT imaging (**left**) and biodistribution studies (**right**) of ^99m^Tc-DPA-ale-Endorem (**A**) and ^99m^Tc-DPA-ale (**B**); the accumulation of both agents in the organs of healthy mice is expressed as the percentage of injected dose per gram of tissue (% ID/g). For each organ, the % ID/g represents the mean value ± standard deviation of *n* = 3 mice (L: liver, S: spleen). (Reprinted with permission from [34]. Copyright 2011 American Chemical Society).

**Figure 2 nanomaterials-13-00503-f002:**
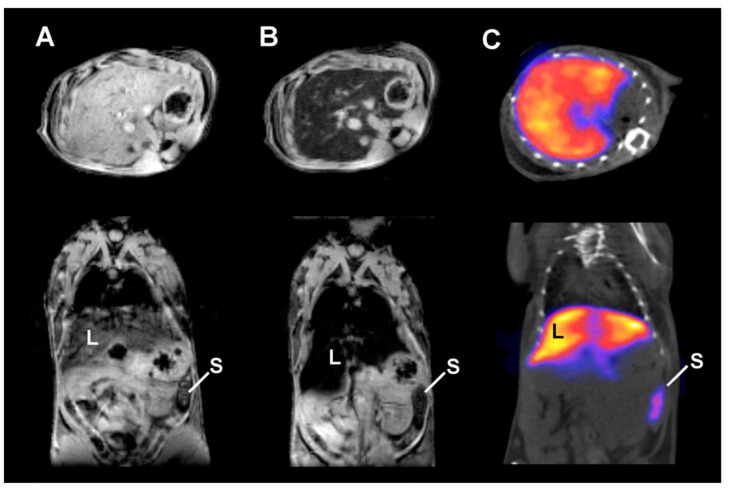
Representative (**A**,**B**) T_2_*-weighted axial (**top**) and coronal (**bottom**) MRI data of ^99m^Tc-DPA-ale-Endorem: (**A**) before injection; (**B**) 15 min post-injection (p.i.); (**C**) Representative SPECT-CT image of the same mice 45 min p.i. (L: liver, S: spleen). (Reprinted with permission from [34]. Copyright 2011 American Chemical Society).

**Figure 3 nanomaterials-13-00503-f003:**
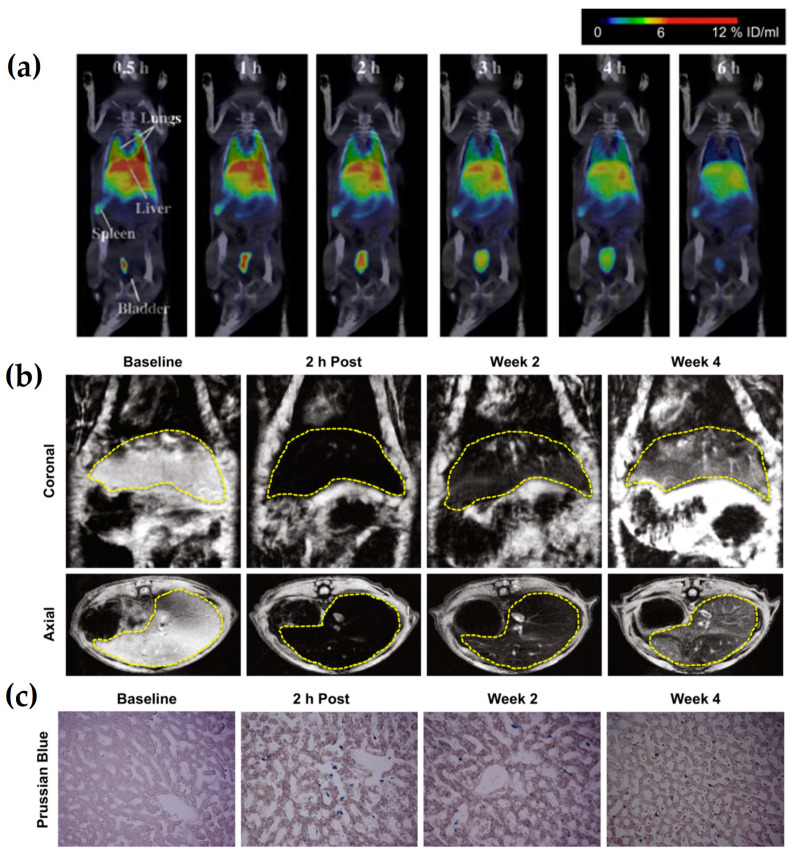
(**a**) Representative SPECT/CT images of a rat at 0.5–6.0 h after injection of ^99m^Tc-CS–IONs; (**b**) T_2_ MR images of rat livers; and (**c**) Prussian blue staining for CS-IONs in the liver before and at 2 h, 2 weeks, and 4 weeks after injection of CS-IONs (Reprinted from [38], with permission from Elsevier).

**Figure 4 nanomaterials-13-00503-f004:**
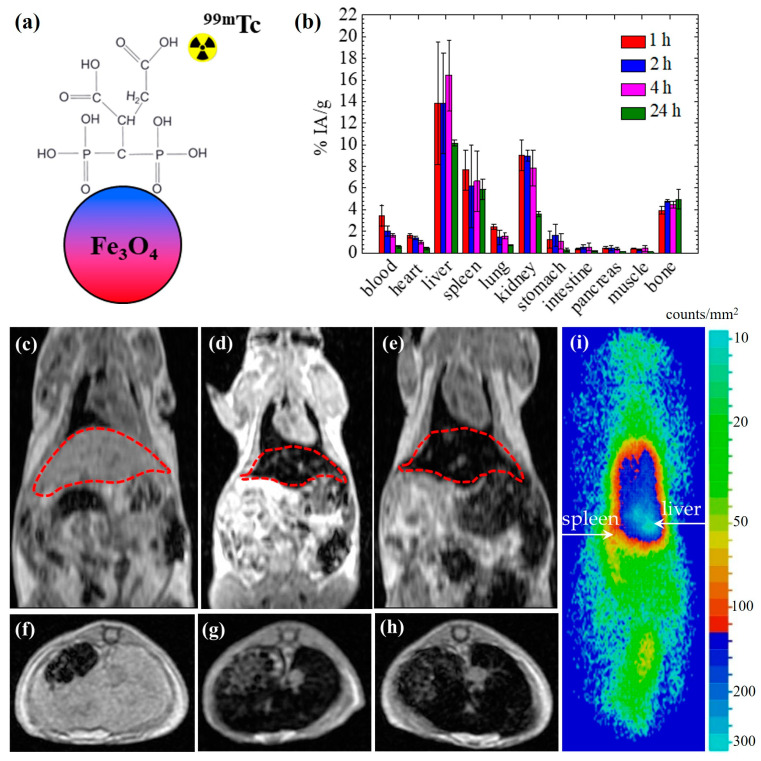
(**a**) The structure of ^99m^Tc radiolabeled and 2,3-dicarboxypropane-1,1-diphosphonic acid coated IONs; (**b**) The biodistribution of RIONs in normal mice at 1, 2, 4, and 24 h p.i.; representative T_1_-weighted (**c**–**e**) coronal and (**f**–**h**) axial MRI data of *n* = 4 normal Swiss mice; (**c**,**f**) without any RIONs/non-radiolabeled IONs; (**d**,**g**) with non-radiolabeled IONs at a concentration of C = 0.1 mg/mL; (**e**,**h**) with RIONs at C = 0.1 mg/mL; the MR imaging was performed at 6 h p.i. The areas of interest, that is, the liver and spleen, are demonstrated by the dotted red lines; (**i**) Representative coronal image of a mouse injected with RIONs obtained at 4 h p.i. by gamma camera [40].

**Figure 5 nanomaterials-13-00503-f005:**
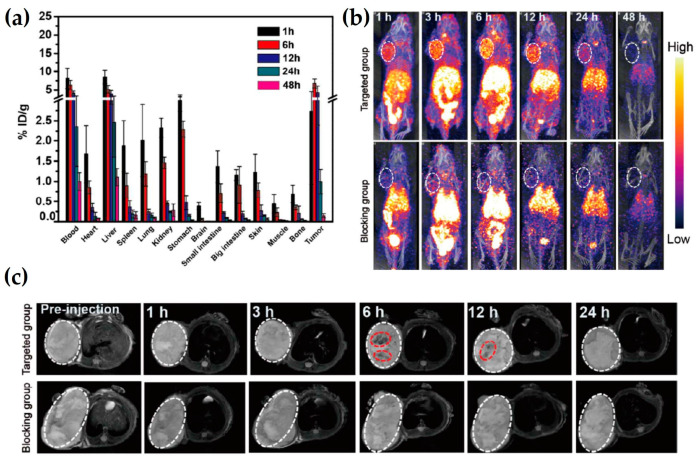
(**a**) The in vivo biodistribution of ^125^I-c(RGDyK)-DMCA performed at 1, 6, 12, 24, and 48 h p.i. The %ID/g corresponds to the percentage of injected dose per gram of tissue; (**b**) Representative SPECT images of tumor-bearing mice which have been obtained up to 48 h p.i. Mice administered with ^125^I-c(RGDyK)-DMCA (targeted group). Mice administered with free RGD, along with ^125^I-c(RGDyK)-DMCA (blocking group); (**c**) Representative T_2_-weighted MR images of tumor-bearing mice which have been obtained up to 24 h p.i. Mice administered with non-radiolabeled c(RGDyK)-IONs (targeted group). Mice administered with free RGD along with c(RGDyK)-IONs (blocking group). The dotted white lines indicate the tumor area and the dotted red lines indicate the presence of c(RGDyK)-IONs (Reprinted with permission from [51]. Copyright 2016 American Chemical Society).

**Figure 6 nanomaterials-13-00503-f006:**
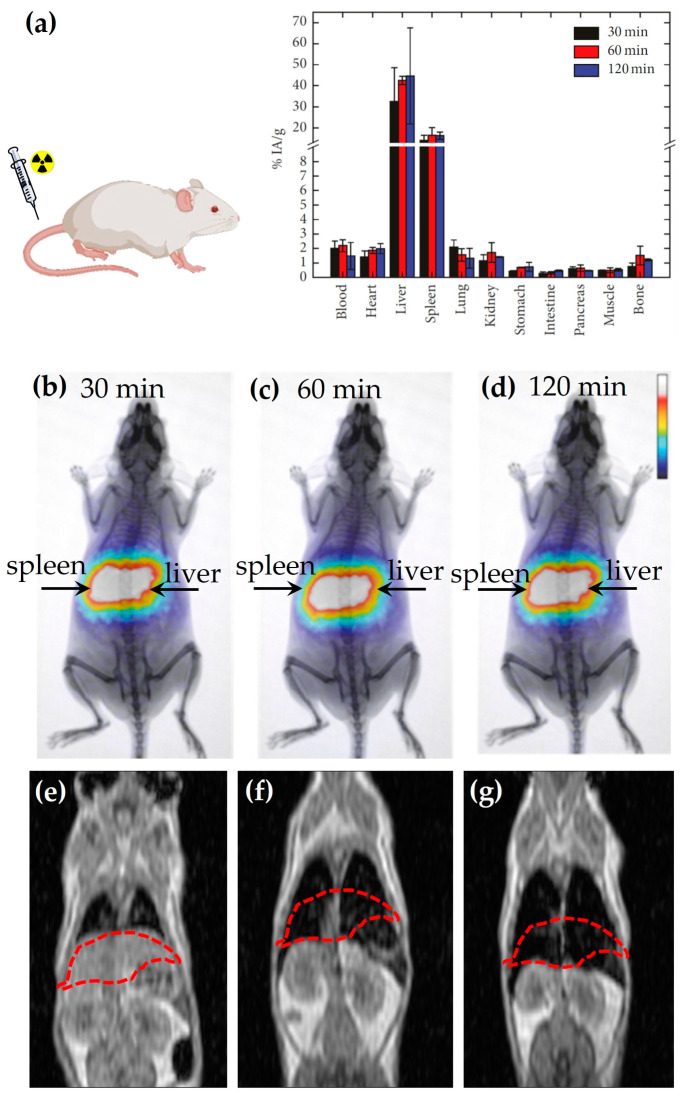
(**a**) In vivo biodistribution study of ^68^Ga based RIONs in normal Swiss mice performed at 30, 60, and 120 min p.i. [58]; (**b**–**d**) Representative images of a mouse obtained by PET imaging at (**b**) 30 min, (**c**) 60 min, (**d**) 120 min p.i. [58]; T_1_-weighted MR image of mice injected via the tail vein with RIONs at a final concentration of (**e**) 0.01 mg/mL; (**f**) 0.1 mg/mL; (**g**) T_1_-weighted MR image of a mouse injected via the tail vein with non-radiolabeled IONs at a final concentration of 0.3 mg/mL. The MRI imaging has been performed at 6 h p.i. [10].

**Figure 7 nanomaterials-13-00503-f007:**
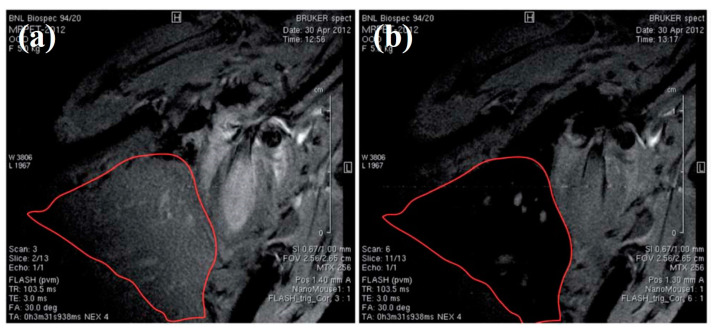
Representative T_2_*-weighted MR images of mouse liver acquired (**a**) before and (**b**) after the administration of non-radiolabeled IONs (Used with permission of Royal Society of Chemistry, from [64]; permission conveyed through Copyright Clearance Center, Inc.).

**Figure 8 nanomaterials-13-00503-f008:**
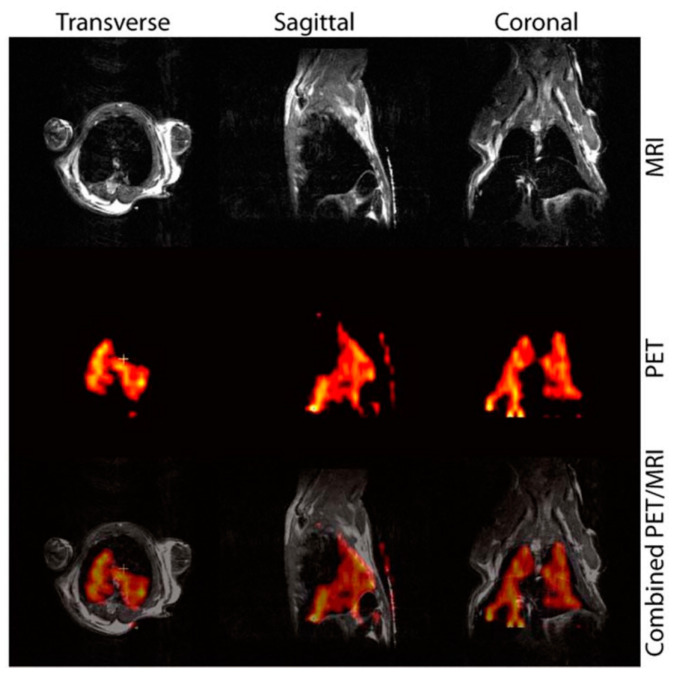
Simultaneous PET/MR images of a mouse, which has been intravenously injected with ^11^C based RIONs; top row: the MR images indicate a significant darkening of the liver due to the presence of ^11^C based RIONs; middle row: the radioactive signal is a result of the decay of ^11^C radionuclide and indicates the presence of ^11^C based RIONs in the liver; bottom row: fused PET/MR images (Used with permission of Royal Society of Chemistry, from [64]; permission conveyed through Copyright Clearance Center, Inc.).

**Figure 9 nanomaterials-13-00503-f009:**
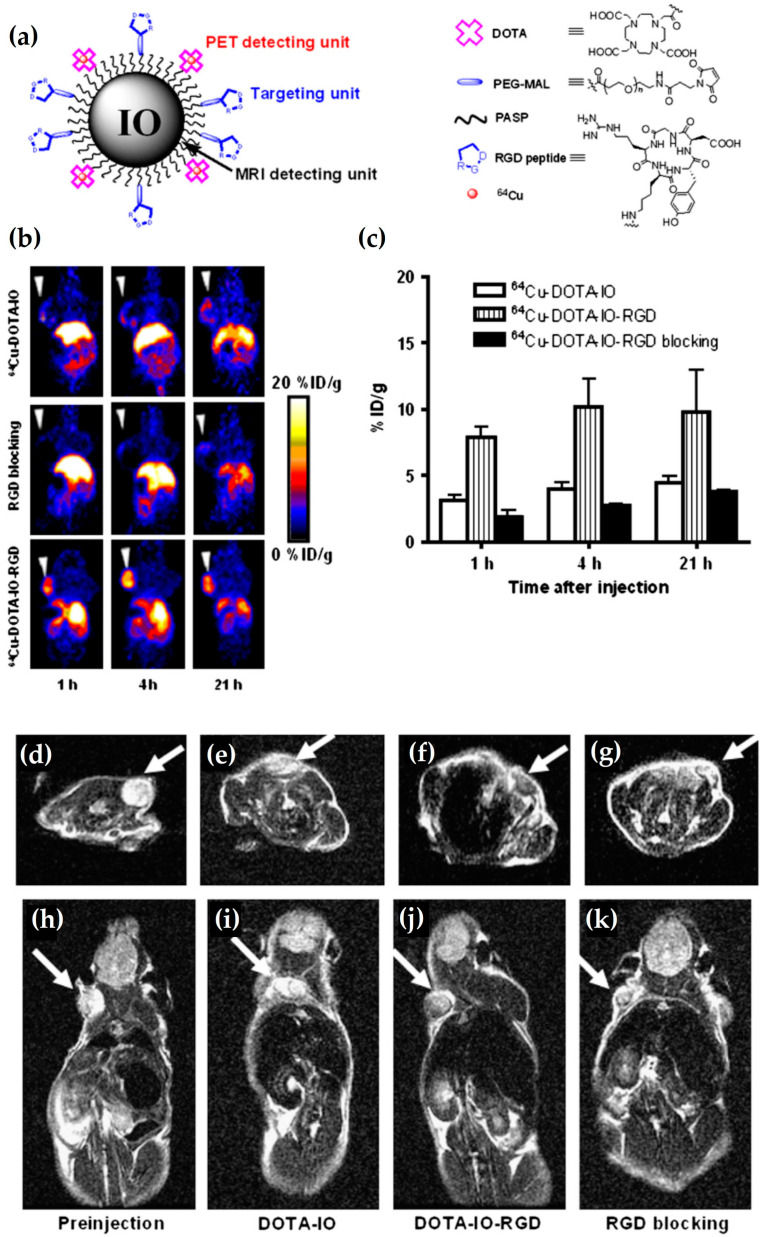
(**a**) Schematic illustration of RIONs, namely ^64^Cu-DOTA-IONs-RGD, for PET/MR imaging; (**b**) Representative coronal PET images of U87MG tumor bearing mice injected with 3.7 MBq of ^64^Cu-DOTA-IONs (top row), ^64^Cu-DOTA-IONs-RGD with 10 mg of c(RGDyK) peptide per kg (middle row) and ^64^Cu-DOTA-IONs-RGD (bottom row) at 1, 4, and 24 h p.i.; (**c**) Time-activity diagram of tumors injected with 3.7 MBq of ^64^Cu-DOTA-IONs, ^64^Cu-DOTA-IONs-RGD, ^64^Cu-DOTA-IONs-RGD with blocking dose of c(RGDyK) peptide; Representative (**d**–**g**) axial and (**h**–**k**) coronal T_2_-weighted MR images of U87MG tumor bearing mice (**d**,**h**) before and 4 h p.i. with (**e**,**i**) non-radiolabeled DOTA-IONs, (**f**,**j**) non-radiolabeled DOTA-IONs-RGD and (**g**,**k**) non-radiolabeled DOTA-IONs-RGD with blocking dose of c(RGDyK) peptide. The U87MG tumor is indicated by the white arrows (This research was originally published in JNM. Lee, H.-Y., Li, Z., Chen, K., Hsu, A.R., Xu, C., Xie, J., Sun, S., Chen, X. PET/MRI Dual-Modality Tumor Imaging Using Arginine-Glycine-Aspartic (RGD)—Conjugated Radiolabeled Iron Oxide Nanoparticles. J. Nucl. Med. 2008; 49(8): 1371–1379 © SNMMI [69]).

**Figure 10 nanomaterials-13-00503-f010:**
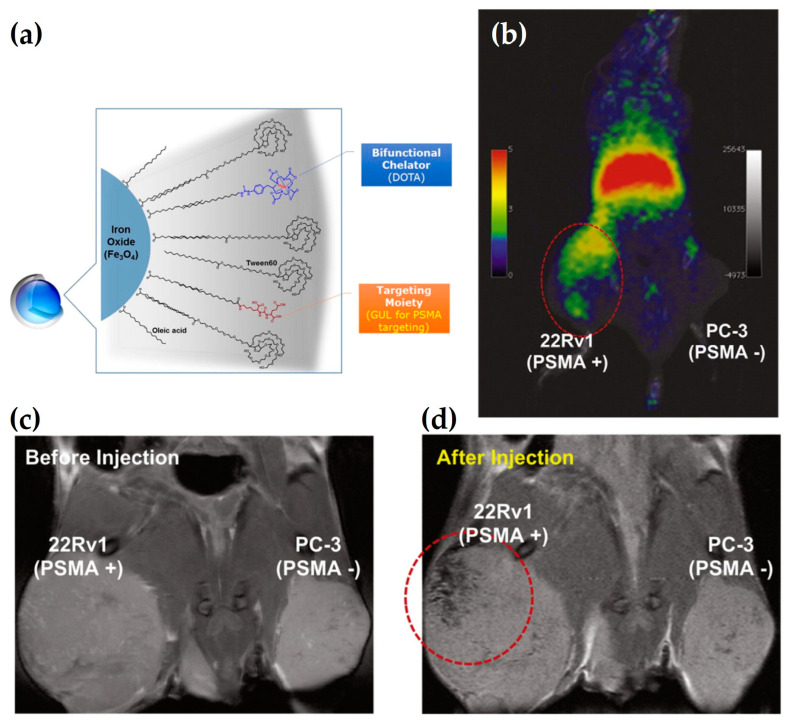
(**a**) The IONs have been radiolabeled with ^68^Ga via the use of DOTA chelator and are conjugated with the targeting ligand, glutamate-urea-lysine (GUL), to selectively target the prostate for PET/MRI dual imaging; (**b**) Representative PET imaging data obtained after 1 h injection of ^68^Ga-DOTA-IONs-GUL DMCA in a BALB/c mouse with PSMA-positive and negative cell lines; Representative T_2_-weighted MR images (**c**) before and (**d**) after 1 h injection of DOTA-IONs-GUL in a BALB/c mouse with PSMA-positive and negative cell lines. The red dotted circles indicate the selective uptake of (**b**) ^68^Ga-DOTA-IONs-GUL and (**d**) DOTA-IONs-GUL, respectively, to the 22Rv1 positive tumor (Reprinted from [72], with permission from Elsevier).

**Table 1 nanomaterials-13-00503-t001:** List of radionuclides used for SPECT imaging [26,27,28].

Radionuclides	T_1/2_ ^1^	Emission	γ-Energy in keV	Generation Mode
^99m^Tc	6 h	γ	140	^99^Mo/^99m^Tc generator
^111^In	67.2 h	Auger, γ	245, 171	Cyclotron
^123^I	13.2 h	Auger, γ	159	Cyclotron
^125^I	60.1 days	γ	36	Cyclotron
^131^Ι	8 days	γ (81.2%), β	284, 364, 637	Cyclotron
^67^Ga	78.3 h	γ	93, 184.6, 300, 393	Cyclotron
^177^Lu	6.7 days	β, γ	208, 113	^177m^Lu/^177^Lu generator

^1^ The half-life of the radionuclide.

**Table 2 nanomaterials-13-00503-t002:** List of radionuclides used for PET imaging [26,27,28].

Radionuclides	T_1/2_ ^1^	Emission	γ-Energy in keV	Generation Mode
^18^F	1.83 h	β^+^	511	Cyclotron
^68^Ga	67.7 min	β^+^	1077	^68^Ge/^68^Ga generator
^64^Cu	12.7 h	β^+^, β	511	Cyclotron
^11^C	20.36 min	β^+^	386 ^2^	Cyclotron
^13^N	9.96 min	β^+^	492 ^1^	Cyclotron
^89^Zr	3.3 d	β^+^	909	Cyclotron

^1^ The half-life of the radionuclide; ^2^ The value corresponds to β^+^ energy in keV.

## Data Availability

Not applicable.

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
