# Peer review of "Radiolabeled Iron Oxide Nanoparticles as Dual Modality Contrast Agents in SPECT/MRI and PET/MRI"

_nanomaterials, 2023, doi:10.3390/nano13030503_

Round 1

Reviewer 1 Report

In this review, the authors report a review on the work published on SPECT/MRI and PET/MRI  dual imaging probes based on radiolabeled Iron oxide nanoparticles (RIONs), working in vivo through passive or selective targeting mechanisms. The work is comprehensive, well organized and provide a clear and informative guide on this subject. However, some aspects are not clear and the conclusions section should be expanded. So, before publication, the authors should respond positively to the following questions:

1) Lines 40-44: the authors state that in the daily clinical practice,  MRI and CT are used to provide information on anatomical abnormalities, whilst
SPECT and PET are able to identify the functional/biochemical changes. While this is normally true today, they should refer that functional MRI is commonly used in clinical studies and hyperpolarized 13C based MRI is already for almost 10 years under clinical trials, eg. for prostate cancer, providing accurate information on biochemical changes. Please discuss in the text.

2) Lines 63 and 88: it is stated correctly that IONPs have been employed as negative MRI contrast agents (CAs) and on lines 99-100, it is stated that IONps of size smaller than 5 nm can be used as positive MRI contrast agents. It would be more informative to say initially that larger (state size) IONPs called SPION are used as negative MRI contrast agents and small ones (called USPION) can be used as positive MRI contrast agents.

3) Lines 93-96: It is true that Gd-based MRI CAs have been involved in NSF bearing patients, and brain deposits of these CAs has been found in patients undergoing multiple contrast enhanced MRI scans, but a) millions of MRI Gd enhanced scans happening worldwide are perfectly safe, in particular the macrocylic ones, where almost no NSF cases have been detected; b) the brain deposits seem to have no harmful effects. So, the Gd-based are not so dangereous as it looks like from the review. Please discuss in the text.

3) On the other hand, the free Fe2+ cations resulting from IONs degradation in the lysosomes, which are not stored into ferritin can be toxic, producing oxygen free radicals via fenton reactions, which can contribute to some IONPs toxicity. Please discuss in the text.

4) Lines 195-204: The authors state that radiolabeling of IONs can be performed via three processes. There can also be used another mechanism involving proton beam activation using  direct irradiation of NPs with protons.

5) Line 329 and Figure 4a: are the NPs stable towards 99mTc release, as it binds only two carboxylate groups?

6) Line 576: In the 68Ga-PEG-IONs how was the 68Ga3+ cation bound to PEG?

7) In section 7 on Selective PET/MRI DMCAs what is the conclusion on the issue of the best chelating agent in terms of thermodynamic and kinetic stability towards 64Cu2+ and 68Ga3+ labeling: DOTA-type or NOTA-type ligands? Please discuss takimg into account cation size and charge versus ligand hole size and flexibility. This could come in the conclusions.

8) Other questions that could be briefly discussed, are: a) best ways to minimize IONs uptake by the RES of liver and spleen; b) passive versus active targeting - are the tumors labeled the same or differently in terms of quantity of NPs and location in the tumor.

9) There are several typos to be corrected: line 206, 99m^Tc; line 325, 111^In; line 332, 125^I; line 339, 177^Lu; line 352, 99m^Tc; line  422, 111^In; line 450, 25^I;  line 514, 64^Cu; line 541, 67^Ga; line 630, 11^C; line 658, 89^Zr; line 672, 18^F; line 684, 69^Ge; line 698, 64^Cu; line 756, 68^Ga

Author Response

We have uploaded our reply to the Reviewer 1

Reviewer 2 Report

In this manuscript authors show the synthesis and in vivo investigation of both biodistribution and imaging efficacy of RIONs as potential SPECT/MRI or PET/MRI DMCAs.The review is well described and show current information in field. In my opinion the manuscript could be accept in present form.

My recommendation is: accept

Regards

Author Response

We have uploaded our reply to thr Reviewer 2
